# Diagnostic value of the antiglycoprotein-2 antibody for Crohn's disease: a PRISMA-compliant systematic review and meta-analysis

Chuiwen Deng,[1] Wenli Li,[2] Jing Li,[3] Shulan Zhang,[1] Yongzhe Li[1]

► Prepublication history and additional material are available. To view these files please visit the journal online (http://dx.doi.org/10.1136/bmjopen-2016-014843).

CD, WL and JL are co-first authors.

## ABSTRACT

**Objectives** To perform a meta-analysis to evaluate the diagnostic performance of the antiglycoprotein-2 (GP2) antibody for Crohn's disease (CD).

**Methods** Three databases (EMBASE, ISI Web of Knowledge and PubMed) were systematically searched. There were 17 eligible studies included in the meta-analysis. A total of 2439 patients with CD and 3184 controls were involved in these studies. STATA V.11.2 and Meta-DiSc V.1.4 were used to perform the meta-analysis.

**Results** The area under the summary receiver operating characteristic curve was 0.68–0.72. The pooled diagnostic sensitivity of the anti-GP2 antibody ranged from 14% to 24%, and the specificity was 96%–98%.

**Conclusions** The anti-GP2 antibody is a specific biomarker for CD, and further exploration of its prevalence among different clinical phenotypes of CD will provide a better understanding of its diagnostic performance.

[1]Department of Rheumatology and Clinical Immunology, Peking Union Medical College Hospital, Chinese Academy of Medical Sciences & Peking Union Medical College, Key Laboratory of Rheumatology and Clinical Immunology, Ministry of Education, Beijing, China
[2]Department of Rheumatology, China-Japan Friendship Hospital, Beijing, China
[3]Clinical Laboratory, Peking University International Hospital, Beijing, China

**Correspondence to**
Dr Yongzhe Li;
yongzhelipumch@126.com

## Strengths and limitations of this study

► This meta-analysis was prepared according to the Preferred Reporting Items for Systematic Review and Meta-Analysis.
► The methods that were applied in this meta-analysis included evaluations of the threshold effect, pooled statistics, operating characteristic curve, publication bias and other sources of heterogeneity.
► The full-text screening, data extraction and analysis of the included studies were conducted by two reviewers independently.
► This is the first systematic review and meta-analysis to evaluate the diagnostic performance of the anti-glycoprotein-2 antibody for Crohn's disease.
► Some of the included studies were of different aims from the meta-analysis for few researches were undertaken to precisely evaluate the diagnostic performance of anti-GP2 antibody, and this strategy may introduce potential heterogeneity.

## INTRODUCTION

Crohn's disease (CD) is a chronic inflammatory disorder of the gastrointestinal tract. Usually, the diagnosis of CD is based on current standard clinical, radiological, endoscopic and histological criteria.[1] However, clinical symptoms and radiological evidence are not specific, while endoscopic and histological tests are invasive procedures that are not always accepted by patients. A precise method for diagnosing CD and differentiating it from ulcerative colitis (UC) is urgently needed.

There is cumulative evidence from cross-sectional and longitudinal studies to support the value of serological markers in the diagnosis, differential diagnosis and prediction of disease course in CD.[2] The most widely used serological marker for CD is the anti-*Saccharomyces cerevisiae* antibody. However, its diagnostic performance in CD has not achieved sufficiently high sensitivity or specificity for clinical needs.[3] Recently, pancreatic autoantibody (PAB) has emerged as a potential diagnostic marker for CD.[4]

The presence of PAB was first identified by indirect immunofluorescence.[5] Recently, the target antigens of PAB have been identified as the pancreatic major granule glycoprotein 2 (GP2) of the zymogen granule membrane and the CUZD1 protein.[4] The anti-GP2 antibody can be detected using an ELISA in routine practice.

Over the past decade, numerous studies have evaluated the anti-GP2 antibody for its ability to accurately diagnose CD. However, inconsistent conclusions relating to the diagnostic performance of anti-GP2 antibody have been drawn.[4] [6–21] The reported diagnostic sensitivity of anti-GP2 antibody (IgG) ranges from 5% to 40%, and the reported diagnostic specificity ranges from 84% to 100%. The IgA subtype of the anti-GP2 antibody is also considered to be potentially valuable in CD diagnosis, but the reported sensitivities (1%–50%) and specificities (84%–100%) also show high variation. In addition, some researchers have proposed that combing

the results of both IgG and IgA subtypes of the anti-GP2 antibody would improve its diagnostic value of CD, while others have disagreed. Lastly, methodological discrepancies, including the subtypes of CD tested, the method of autoantibody detection and the manufacturer of the detection kits might influence the diagnostic value of the anti-GP2 antibody, and these should be investigated.

In order to verify the diagnostic performance of the anti-GP2 antibody in patients with CD and to determine the factors that influence the results of anti-GP2 antibody testing, we performed the present systematic review and meta-analysis

## METHODS

### Literature search

Studies were identified in EMBASE, ISI Web of Knowledge and PubMed databases. To retrieve all relevant publications related to anti-GP2 antibody and CD, we searched for the follow terms: 'anti-glycoprotein 2 antibody', 'glycoprotein 2 autoantibodies' and 'autoantibodies to glycoprotein 2', combined with 'Crohn's disease' and 'CD'. No limits were placed on ethnicity or geographic region, and all documents were included up to June 2016. The specific PubMed search algorithm is provided in online supplementary file 1. Additional relevant references cited in searched articles were also selected, if any. All analyses in this systemic review were based on previously published studies, and thus no ethical approval or patient consent was required.

### Eligibility criteria

The following criteria were used to determine eligibility for inclusion: (1) studies that assessed the diagnostic accuracy of the anti-GP2 antibody for CD were included; (2) studies with sufficient data to construct two-by-two tables were included; (3) the control groups set for the CD patients should fulfil one of the following: people with complaints (abdominal pain, diarrhoea, ileus, and so on) that made CD a relevant diagnostic possibility and patients diagnosed with the diseases that need to be differential with CD, such as UC, irritable bowl syndrome, ischaemic bowel disease and so on; (4) all published languages were included; and (5) studies based on animals or cell cultures or case reports without subsequent publication in full text were excluded. In the case of overlapping studies, only the study with the largest sample size was included in our analysis.

### Data extraction

Data were extracted from all selected studies by two independent investigators. Inter-researcher disagreements were resolved by consensus or by a third investigator. The following data were collected from each selected study: first author's name, publication year, country in which the study was performed and study results. Formally, the combined test of IgG and IgA subtypes of the anti-GP2 antibody is taken as positive if one or both tests is positive (logical OR of positive results); it is negative if both tests are negative (logical AND of negativity). Basing on this knowledge, the 'either IgG or IgA' phrase of this meta-analysis means that either IgG or IgA subtypes of the anti-GP2 antibody is found to be present (abnormal). Study quality was assessed using the Quality Assessment of Diagnostic Accuracy Studies (QUADAS) tool. Authors of the identified studies were contacted via email if further study details were needed.

### Statistical analysis

Statistical analysis was performed using STATA V.11.2 and Meta-DiSc V.1.4 (Unit of Clinical Biostatistics, Ramon y Cajal Hospital, Madrid, Spain). Potential important differences in the results of the individual studies are frequently referred to as heterogeneity. In this meta-analysis, the heterogeneity among studies was evaluated by Cochrane's $Q$-statistic as well as by the $I^2$-statistic. A $Q$-statistic p value >0.10 indicated lack of heterogeneity among studies. $I^2$ <25% was considered low heterogeneity, 25%–50% moderate and >50% indicated a high degree of heterogeneity. Finally, the overall or pooled sensitivity, specificity, positive likelihood ratio (LR+), negative likelihood ratio (LR–) and their 95% CIs were obtained by a random-effects or a fixed-effects model in the presence (p≤0.10 or $I^2$ >50%) or absence (p>0.10 and $I^2$ ≤50%) of heterogeneity, respectively. The area under the summary receiver operating characteristic (SROC) curve represented the overall performance of the detection method. Besides the area under the curve (AUC), the ROC analysis also provide estimates of uncertainty that includes a 95% confidence region that indicates the precision of studies in the pooled estimate, and a 95% prediction region around the summary points that illustrate the amount of between study variation. A p value of <0.05 (two sided) was considered significant. Threshold effects occur when different cut-off values are applied in different studies, leading to heterogeneity of the results. A Spearman's rank correlation calculating between the logarithm of sensitivity and the logarithm of (1 – specificity) across source studies was performed to test for threshold effects. Evaluation of publication bias was also undertaken.

## RESULTS

### Literature search

Electronic and manual searches yielded a total of 323 potentially eligible articles. A flow chart of the screening of articles for meta-analysis is illustrated in figure 1. There were 287 articles that were excluded based on screening of the titles and abstracts. A further 19 full-text articles were excluded as they were not related to our subject. Finally, a total of 17 eligible studies were included in the meta-analysis.[4 6–21]

### Study characteristics

The characteristics of the 17 studies are summarised in online supplementary table 1. A total of 2439 CD patients and 3184 controls were involved in these studies. With

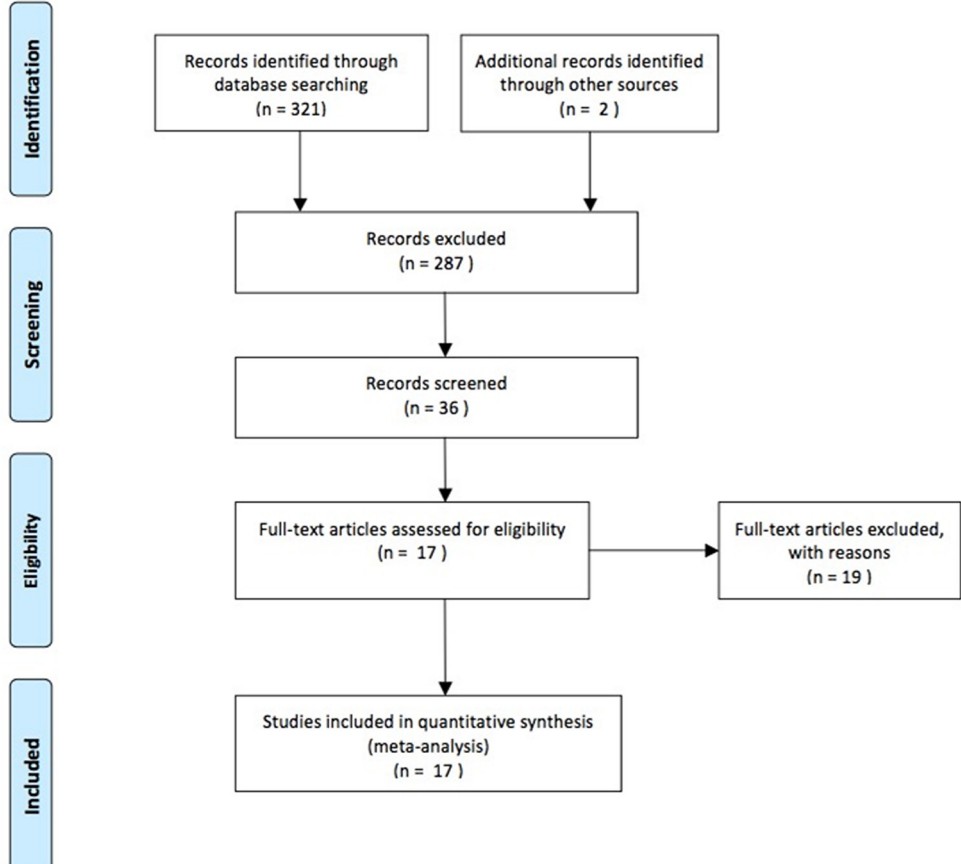

**Figure 1** Flow chart of studies included in the meta-analysis.

regard to the geographic location of the studies, seven were carried out in Germany,[4 7 8 11 15 18 21] four in the UK[6 10 17 20] and one each in Serbia, the Netherlands, Hungary, the Czech Republic, Belgium and China.[9 12–14 16 19] Assessment using QUADAS indicated that the studies were of median to high quality, with positive results for at least 6/14 items.

### Meta-analysis of the diagnostic accuracy of the anti-GP2 antibody (IgG) for CD

A total of 2439 CD patients and 3191 controls were involved in this part of the meta-analysis. The diagnostic sensitivity of the anti-GP2 antibody (IgG) for CD ranged from 5%–38%, and the reported specificity ranged from 84% to 100%.[4 6–21] The diagnostic OR was 7 (95% CI 5 to 11; $Q$=87.68, p<0.01; $I^2$=98%). The pooled sensitivity was 19% (95% CI 14% to 25%; $Q$=179.16, p<0.01; $I^2$=91%), and the pooled specificity was 97% (95% CI 94% to 98%; $Q$=75.97, p<0.01; $I^2$=78%). The LR+ and LR− were 6.1 (95% CI 4.1 to 9.1) and 0.84 (95% CI 0.79 to 0.89), respectively. The AUC of the SROC was 0.71 (95% CI 0.67 to 0.75). The 95% confidence region of the SROC was narrow and small, increasing the precision of studies in the pooled estimate. The 95% prediction region of the SROC was broad and large, suggesting heterogeneity between studies. The sensitivity forest plots, specificity forest plots and SROC are shown in figure 2A, B and C, respectively.

### Meta-analysis of the diagnostic accuracy of the anti-GP2 antibody (IgA) for CD

A total of 2214 CD patients and 2894 controls were involved in this part of the meta-analysis. The diagnostic sensitivity of the anti-GP2 antibody (IgA) for CD ranged from 1%–50%, and the reported specificity ranged from 84% to 100%.[4 6–16 18–21] The pooled sensitivity was 14% (95% CI 9% to 20%; $Q$=213.03, p<0.01; $I^2$=93%), and the pooled specificity was 98% (95% CI 96% to 99%; $Q$=156.06, p<0.01; $I^2$=90%). The LR+ and LR− were 6.3 (95% CI 4.1 to 9.7) and 0.88 (95% CI 0.84 to 0.93), respectively. The AUC of the SROC was 0.68 (95% CI 0.64 to 0.72). The 95% confidence region of the SROC was narrow and small, increasing the precision of studies in the pooled estimate. The 95% prediction region of the SROC was broad and large, suggesting heterogeneity between studies. The sensitivity forest plots, specificity forest plots, and SROC are shown in figure 3A, B and C, respectively.

### Meta-analysis of the diagnostic accuracy of the anti-GP2 antibody (either IgG or IgA) for CD

A total of 1818 CD patients and 2195 controls were involved in this part of the meta-analysis. The diagnostic sensitivity of the anti-GP2 antibody (IgG or IgA) for CD ranged from 10% to 54%, and the reported specificity ranged from 84% to 100%.[4 6–12 18 19] The pooled sensitivity

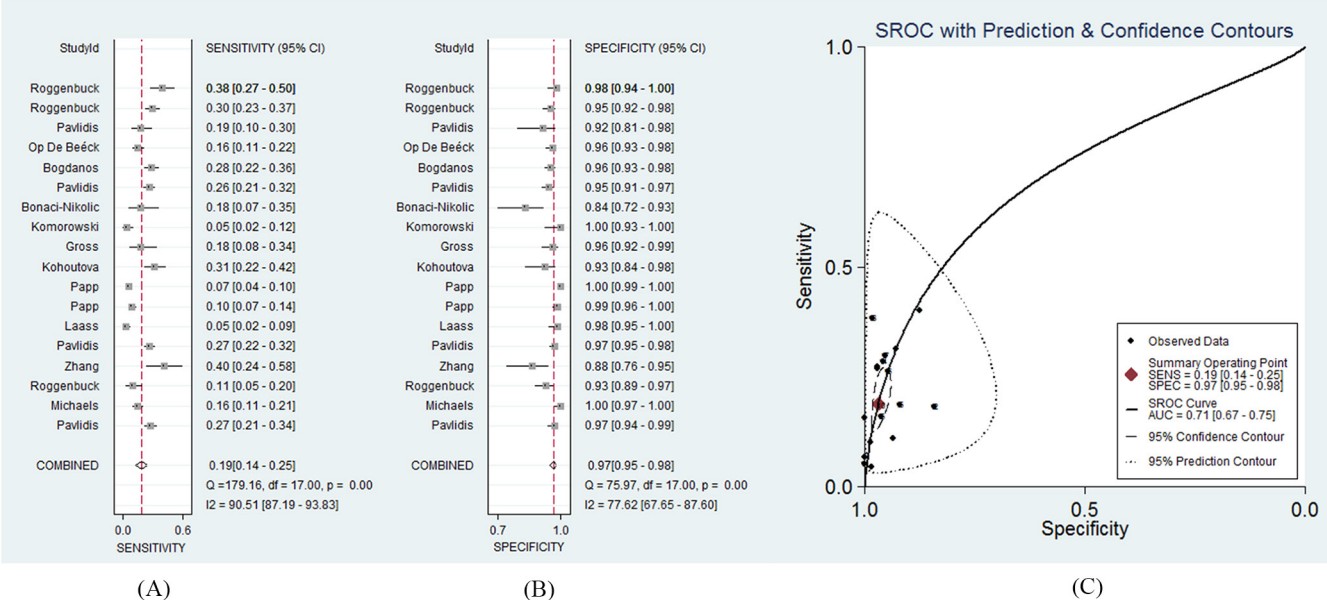

**Figure 2** The forest plots and the summary receiver operating characteristic curves of antiglycoprotein 2 antibody (IgG) for CD. (A) Sensitivity forest plot; (B) specificity forest plot; (C) summary receiver operating characteristic (SROC).

was 24% (95% CI 18% to 32%; $Q$=103.59, p<0.01; $I^2$=90%), and the pooled specificity was 96% (95% CI 93% to 97%; $Q$=55.87, p<0.01; $I^2$=82%). The LR+ and LR− were 5.4 (95% CI 4.1 to 7.2) and 0.79 (95% CI 0.74 to 0.85), respectively. The AUC of the SROC was 0.72 (95% CI 0.68 to 0.76). The 95% confidence region of the SROC was narrow and small, increasing the precision of studies in the pooled estimate. The 95% prediction region of the SROC was broad and large, suggesting heterogeneity between studies. The sensitivity forest plots, specificity

forest plots, and SROC are shown in figure 4A, B and C, respectively.

### Multiple regression analysis and exploration of threshold effect

Meta-regression analysis was conducted to explore possible sources of heterogeneity. The following covariates were evaluated: the method of autoantibody detection (indirect immunofluorescence and ELISA) and the manufacturer of detection kits (Euroimmune, GA Generic, Inova

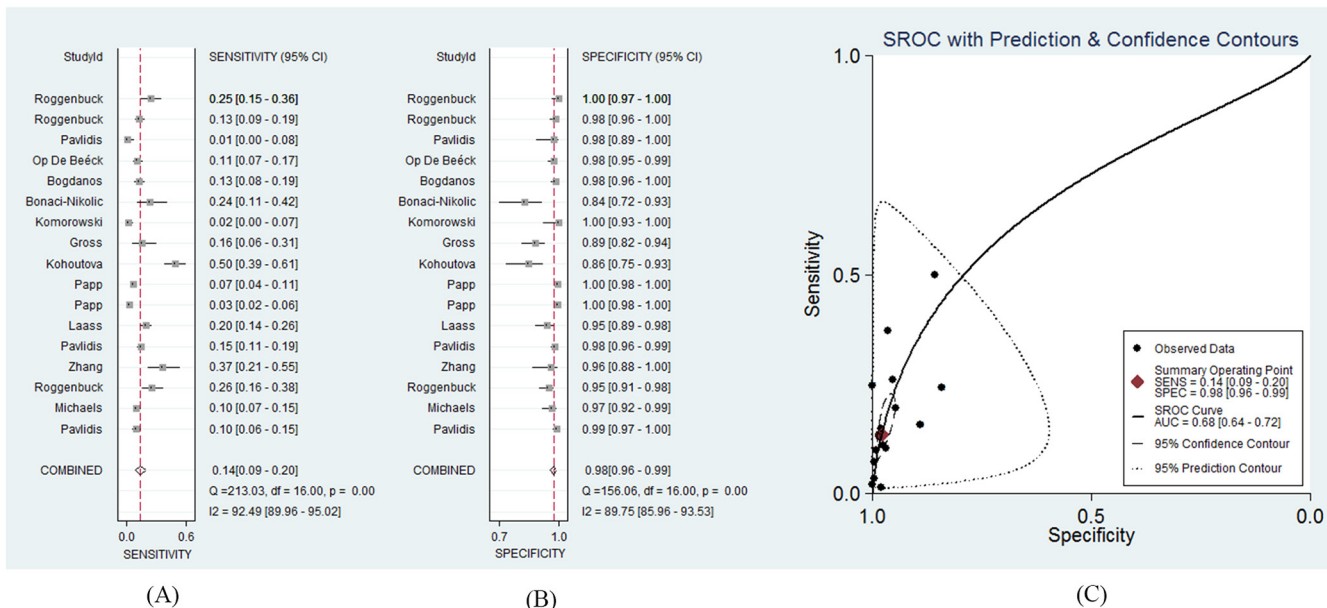

**Figure 3** The forest plots and the summary receiver operating characteristic curves of antiglycoprotein 2 antibody (IgA) for CD. (A) Sensitivity forest plot; (B) specificity forest plot; (C) summary receiver operating characteristic (SROC). CD, Crohn's disease.

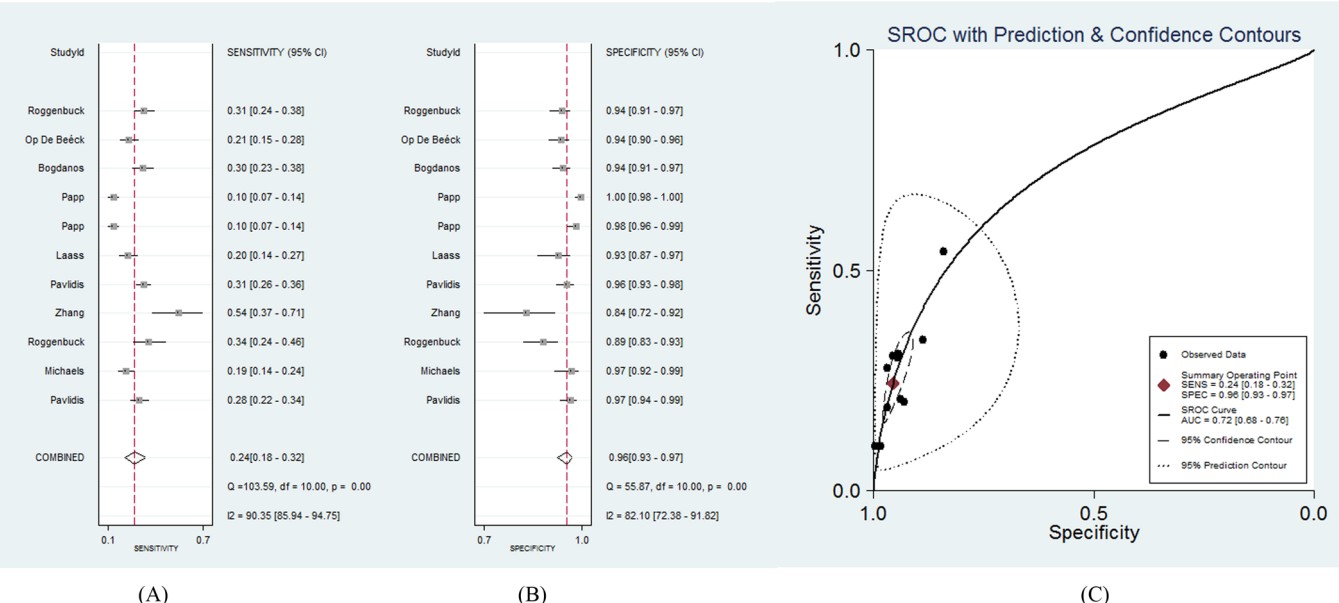

**Figure 4** The forest plots and the summary receiver operating characteristic curves of antiglycoprotein 2 antibody (either IgG or IgA) for CD. (A) Sensitivity forest plot; (B) Specificity forest plot; (C) Summary receiver iperating characteristic (SROC). CD, Crohn's disease .

Diagnostics or in-house kits), the region where the study was performed and the QUADAS scores. Results indicated that the method of autoantibody detection led to heterogeneity. No other sources of heterogeneity were found.

The Spearman's correlation coefficient for anti-GP2 antibody (IgG), anti-GP2 antibody (IgA) and anti-GP2 antibody (either IgG or IgA) were 0.525 (p=0.025), 0.622 (p=0.008), 0.761 (p=0.007), respectively. Basing on these characteristics, a fairly strong and significant correlation is found in each of the three cases, indicating existence of threshold effect.

### Publication bias

The presence of a statistically significant slope coefficient (p<0.05) is believed to indicate possible bias. We conducted funnel plots that represented a somewhat symmetric curve (figure 5). The p-values of the slope coefficients for the anti-GP2 antibody using IgG, IgA, and either IgG or IgA were determined to be 0.11, 0.10, and 0.22, respectively, indicating that no publication bias was observed among the included studies.

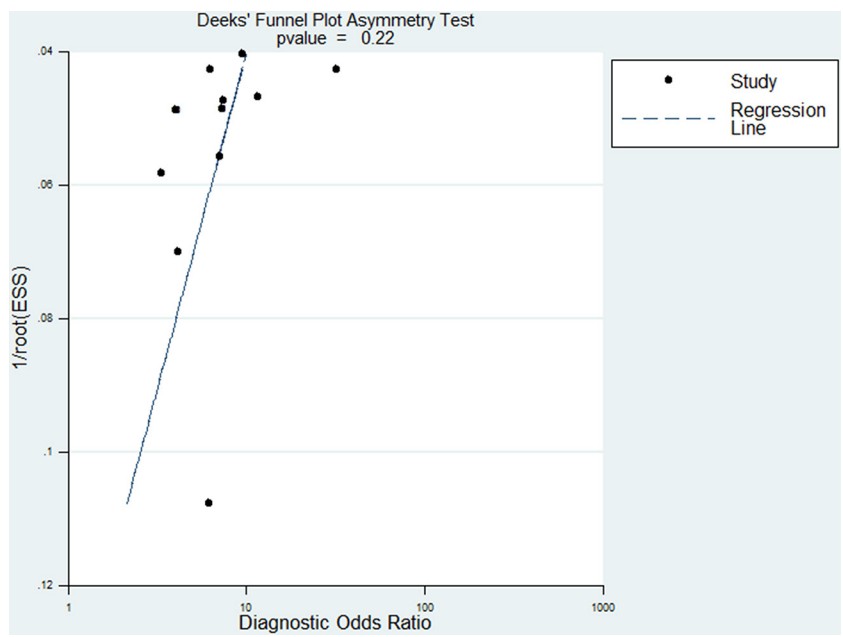

**Figure 5** Funnel plot of the publication bias for the antiglycoprotein 2 antibody (IgG and IgA either). ESS is the abbreviation of effective sample size.

## DISCUSSION

The diagnosis of CD currently relies on a triad of clinical characteristics: radiological features and endoscopic and histological assessment. However, common clinical manifestations are not specific to CD. Furthermore, endoscopic and histological assessments are invasive and may not be easily tolerated by patients and clinicians. Non-invasive assessments alone, such as radiological tests, may not be adequate for diagnosing CD. To date, anti-GP2 antibody is considering an emerging factor for diagnosis of CD. To our knowledge, this study is the first meta-analysis to provide precise and controlled data on the diagnostic performance of the anti-GP2 antibody in CD.

There were 17 studies of high quality that were included in this meta-analysis (see online supplementary table 1). To explore the diagnostic performance of the anti-GP2 antibody IgG subtype, IgA subtype and either the IgG or IgA subtype, we calculated statistics for each of these.

Generally, studies can be pooled together to calculate pooled statistics in diagnostic meta-analysis when there is no threshold effect. If not, the AUC of the SROCs should be calculated instead. Results of our study suggest that there were fairly strong and significant threshold effects in the recruited studies. Thus, we calculated the AUC of the SROCs. The pooled statistics were also calculated for reference.

For the anti-GP2 antibody IgG subtype, the AUC (0.71) of the SROC implies that the diagnostic performance of the anti-GP2 antibody IgG subtype is not satisfactory enough for CD. The pooled specificity of the anti-GP2 antibody IgG subtype was high (97%), with a relatively low sensitivity (19%). Compared with the IgG subtype, the anti-GP2 antibody IgA subtype showed a smaller AUC of the SROC (0.68), higher specificity (98%) and lower sensitivity (14%). Considering both the IgG and IgA subtype in the CD diagnosis, the pooled sensitivity and the AUC of the SROC was higher (24% and 0.72), while the pooled specificity was lower (96%) than when only one subtype was used.

As we previously said that CD symptoms and manifestations are all non-specific. In other words, a specific test is needed for CD diagnosis. The results mentioned above showed that anti-GP2 antibody was very specific for CD. Especially in the high specificity–low sensitivity situation we are dealing with, CD is more reliably identified than controls by anti-GP2 antibody. Admittedly, this occurs at the price of allowing identification of only a small fraction of CD cases because the majority are false negatives.

In addition, these results show that testing for both the IgG and IgA subtypes of the anti-GP2 antibody may result in better diagnostic performance for CD than either the IgA or IgG subtype alone. However, high levels of heterogeneity were detected in among the prediction regions of the SROC, pooled sensitivity and specificity of the anti-GP2 antibody when testing for both the IgG and IgA subtypes. Therefore, we conducted a meta-regression analysis for the anti-GP2 antibody. However, no significant heterogeneity was found among the manufacturer of

detection kits, the region where the study was performed or the QUADAS scores when testing for both the IgG and IgA subtypes. Interestingly, the method of autoantibody detection was confirmed to be the source of the heterogeneity in the diagnostic specificity of the anti-GP2 antibody (IgG) in CD.

Clinical phenotypes of CD patients are determined based on the Montreal Classification.[22] Specifically, CD is described by A, L and B classifications. The A classification represents the age at diagnosis (A1, <17 year; A2, 17–40 year; A3, >40 year), L represents the location of the disease (L1, ileal; L2, colonic; L3, ileocolonic; L4, upper gastrointestinal tract) and B represents the disease behaviour (B1, non-stricturing, non-penetrating; B2, stricturing; B3, penetrating; P, perianal disease modifier). Recently, an association between the loss of tolerance to GP2 and the phenotype of CD disease in accordance with the Montreal classification has been reported. The anti-GP2 antibody was found to be more prevalent among CD patients with the A1, B2, L1 and L3 phenotypes when compared with those with the A2, A3, B1, B3 and L2 phenotypes.[6 7 12 17 18] Differences among the phenotypes of CD patients recruited in the eligible studies might result in heterogeneity in our meta-analysis. However, most of the recruited studies did not provide detection ratios of anti-GP2 antibody among the different subtypes of CD, so we are unable to consider this in our meta-analysis.

Of note, the meta-analysis revealed LR+ of above 5 for the anti-GP2 antibody. This means that the anti-GP2 antibody exhibits at least a 30% change in probability and, thus, at least a moderate effect on the post-test probability of CD. Thus, the anti-GP2 antibody can be used for the differentiation of CD patients from controls.

Some relevant unpublished studies with negative findings that meet our inclusion criteria might have been missed in our analysis. Therefore, funnel plots were used to detect possible publication bias, And no publication bias was detected in the funnel plots.

There are two limitations of this meta-analysis. The PRISMA checklist assumes that the source studies should have the same aim as the meta-analysis. In this research, some of the included studies were investigations with different aims. Three of them concern subtypes of CD,[7 13 18] two have attention on refractory cases[14] or development of tolerance[11] and three focus on molecular processes.[15 17 21] Some studies may be published for disappointing results or other reasons.[10 16] Up to date, few researches were undertaken precisely to confirm the diagnostic value of anti-GP2 antibody, so we selected the current studies that help us evaluate this clinical index with potential value earlier, and this strategy may introduce potential heterogeneity. The other limitation of this meta-analysis is that we did not compare the diagnostic accuracy of the anti-GP2 antibody with that of the anti-*S. cerevisiae* antibody owing to a lack of relevant studies.

In conclusion, the anti-GP2 antibody is a specific marker of CD, and it can be used for the differentiation of CD patients from controls. However, the diagnostic

performance of the anti-GP2 antibody cannot be definitively concluded owing to heterogeneity among studies that has not been fully explained. Further exploration of the prevalence of the anti-GP2 antibody among different clinical phenotypes of CD will provide a better understanding of its diagnostic performance.

**Contributors** YL designed the research; CD, LW and JL collected and analysis the data; CC wrote the paper; and YL and SZ helped optimise the research and proofread the paper.

**Funding** This work was supported by funding from the National Natural Science Foundation of China Grants (81302591, 81601833, 81501841, 81172857, 81373188), the Research Special Fund for Public Welfare Industry of Health (201202004), the Chinese National High Technology Research and Development Program, Ministry of Science and Technology Grants (2011AA02A113) and the National Science Technology Pillar Program in the 12th Five-year Plan (2014BAI07B00).

**Competing interests** None declared.

**Patient consent** Not applicable.

**Provenance and peer review** Not commissioned; externally peer reviewed.

**Data sharing statement** All raw data are supplied in the supplementary table 1. All recorded information from the data extraction process, not included in the systematic review article, will be available on request.

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
