## [Reviewer comments · BMJ Open]

ARTICLE DETAILS

TITLE (PROVISIONAL)	Diagnostic value of the anti-glycoprotein-2 antibody for Crohn's disease: A PRISMA-compliant systematic review and meta-analysis
AUTHORS	Deng, Chuiwen; Li, Wenli; Li, Jing; Zhang, Shulan; Li, Yongzhe

VERSION 1 - REVIEW

REVIEWER	Sumathi Sankaran-Walters University of California, Davis USA
REVIEW RETURNED	18-Nov-2016

GENERAL COMMENTS	The importance of this study is not stated. Why was this study embarked upon? A paragraph discussing in detail what is know would be useful.
--

REVIEWER	Dirk Roggenbuck Brandenburg University of Technology Institute of Biotechnology Germany Shareholder of Medipan and GA Generic Assays GmbH
REVIEW RETURNED	18-Nov-2016

GENERAL COMMENTS	Deng et al. report an interesting meta-analysis on the diagnostic value of autoantibodies to the major zymogen granule glycoprotein 2 (anti-GP2) in patients with Crohn'S disease (CD). Indeed, the relevance of anti-GP2 for the differential serological diagnosis of IBD has been investigated thoroughly lately and has been in the focus of laboratory diagnosticians. The meta-analysis is well planned and conducted by including 17 studies meeting the eligibility criteria. The discussion mentions potential limitations of this meta-analysis. However, the conclusions are not supported by the data described. Major points: 1. The conclusion of the meta-analysis should be revised. The novel anti-GP2 was reported as a specific marker of CD recently, which was corroborated by this meta-analysis. Anti-GP2 is associated with the clinical phenotype of CD (stricturing/stenosing) or even a putative subtype of CD. Thus, the sensitivity of around 20% cannot be used to postulate that anti-GP2 is "not a useful diagnostic marker for CD".2. The association of the loss of tolerance to GP2 with the phenotype of disease in accordance with the Montreal classification should be mentioned in the INTRODUCTION section and discussed
---

	in the DISCUSSION section in line with the obtained meta-analysis data. 3. The possibility of the existence of different subtypes of CD should be elaborated in the DISCUSSION section in addition to the previous comment. 4. The area under the curve (AUC) of a receiver operating characteristic (ROC) analysis indicates the probability that a randomly selected individual from the positive group has a test result indicating greater suspicion than that for a randomly chosen individual from the negative group. Thus, it is a measure of the assay accuracy. When the variable under study cannot distinguish between the two groups, i.e. where there is no difference between the two distributions, the area will be equal to 0.5. However, the AUC values of the summary ROC obtained in the paper seem to be higher than 0.5. Thus, the significance of the difference from 0.5 should be calculated for each AUC value given in the manuscript. Further, you cannot conclude that an AUC of 0.72 renders the parameter under investigation not useful unless the difference to 0.5 is not significant. 5. The meta-analysis revealed positive likelihood ratios of above 5 for anti-GP2. This means, that anti-GP2 has at least a 30% change in probability and, thus, at least a moderate effect on posttest probability of CD. Thus, anti-GP2 can be used for the differentiation of CD patients from controls, which should be concluded in the DISCUSSION section. Minor:  1. ABSTRACT section, lines 12 - 17: "There were .." – This sentence needs revision on grammar. 2. RESULTS/MULTIPLE ... section, lines 40 – 49: The whole paragraph is difficult to comprehend and needs revision. 3. DISCUSSION section, line 21: Not all diagnostic methods indicated are invasive. Revise the sentence and correct the grammar. 3. DISCUSSION section, line 35: Use the term "diagnostic performance"! 4. Fig. 2, 3, 4 legends: Indicate the full term for SROC. 5. Fig. 5: Introduce the abbreviation "ESS" in the legend. 6. Suppl. Table 1: Correct "anti-granule protein 2" in the headline.
--	--

REVIEWER	Jørgen Hilden Biostatistics, Univ. of Copenhagen, Denmark
REVIEW RETURNED	07-Dec-2016

GENERAL COMMENTS	The Editors should make sure that figures are readable despite the small print! Authors - please see below. December 2016 Review of Deng, Li & al., "Diagnostic value of anti-glycoprotein-2 antibody for Crohn's disease ..." BMJ-open-2016-014843 This is a carefully prepared diagnostic systematic review & meta-analysis about the question in the title, using standard methodology and software. My criticism, as you will see, is mostly concerned with
--

its clinicometric aspects (do the methods and the presentation match clinical realities, quantitatively as well as qualitatively?).

The Editors should make sure that figures are readable despite the small print!

References are to px/y meaning authors' page no. x (= pdf page no. x+1), pdf line no. y.

Minor points

M1) P7/7, "(IgA)": here "(either antibody)" is meant.

M2) It should be stressed from the outset what is meant by 'either' in the phrase 'either IgG or IgA.' Without studying the Supplem. Table 1 I would not have been able to guess the answer. It means that 'either IgG or IgA is detected.' Formally, we here have a combined test that is taken as positive if one or both component tests is positive (logical OR of positive results); it is negative if both components are negative (logical AND of negativity). [There are 16 possible ways of combining two binary tests, including trivial ones such as 'always positive,' and this is just one of the 10 non-trivial ones.]

M3) Similarly, at p15/51 we are told that there are 17 source studies. Without the table readers will wonder why there are 18, 17 and 11 entries in Figs. 2, 3 and 4, respectively. This requires a brief explanation.

M4) Ref. [4] is identical to ref. [21].

M5) Fig. 5 is a funnel plot (although without showing any funnel) 'exploring the possibility' of publication bias. (The plot is not a *picture of a bias* but a tool used to check *whether there is a bias*.) Please rephrase.

M6) On p9/18 the positivity criterion of my M2 above (logical OR of positivity) is misrepresented. In fact, the phrase 'detecting both the IgG and IgA subtype' suggests that a logical AND is employed (combined test positive only if both subtypes are detected). So 'detecting' should be 'looking for' or something similar.

M7) On p9/15&52 (and similarly in the Abstract) 'extremely' is an exaggeration, in my opinion (esp. in view of C3 below).

Statistics

S1) A Spearman correlation (p7/40, p9/37) is calculated but the reader is not told between which variables. I presume it as a correlation between *sens* and $(1 - spec)$ across source studies. Please clarify – and, if expedient, give a precise reference to textbook / manual.

S2) A fairly strong and significant correlation is found in each of the three cases. To me this looks as evidence of a threshold or

threshold-imitating (e.g., severity or chronicity) effect. Why then the negative conclusion? Again clarification is needed.

S3) P8, top: There is circumstantial indications that publication biases may be present (see C2 below). However that may be, the non-significant *P*-values should not be taken as evidence of absence of such bias. (As a general rule, when you test for an effect of some kind, remember that a large *P*-value is not *evidence of absence of effect* but represents *absence of (firm) evidence* about the suspected effect and, and in particular, about its direction.) Please write ‘...respectively, so absence of publication bias cannot be rejected’ or ‘so publication bias has not been demonstrated.’

S4) One piece of criticism falls back on the software people: I couldn't very well read the numbers in Figs. 2-4. However, isn't it true that the meta-analytic result shown in part C of the figures is not included in the text but only given via a diamond in the graph (with no mention of its coordinates)? The text as I read it only mentions the naïve separate meta-analysis of *sens* and *spec* (parts A and B, resp.), which has never been recommended, in my opinion. That is, the more appropriate analysis does not show up in the text, the less appropriate does.

S5) Does the prediction region (outer curve) of part C require a small explanation in order to distinguish it from the confidence region for the central point?

Clinicometrics

C1) I am aware that the authors have had to make do with the data they have been able to find in the world literature. And I appreciate that they have followed the PRISMA guideline. However, in one respect, their text can be criticized for not taking PRISMA seriously: The PRISMA checklist assumes that the studies were undertaken precisely in order to describe a specified diagnostic task and measure the associated accuracy of one or more diagnostic tests, i.e., just as is the aim of the meta-analysis. However, going through the list of refs. [6-22], I find many titles that suggest that the diagnostic 2-by-2 table was just a by-product of an investigation with a different aim. Such datasets are possibly not very concerned about representative samples, etc., and have been subject to selective forces in the publication process that may bias the meta-analysis in unpredictable ways. To be specific, three titles concern *subtypes* of CD [7, 13, 18]; two primarily deal with *refractory cases* [14] or *development of tolerance* [11]; three appear to focus on *molecular processes* [15, 17, 22], and one advertises a *new kit* [10]. Novelty is also a theme in [21] ('exciting-news bias?'), and the purpose of [16] is to warn against a diagnostic 'trap' ('disappointing-news bias?'). I would wish that one were able to add a few words to Eligibility Criterion (1) on p4/29, writing (instead of '[studies] that assessed the diagnostic accuracy of ...'): 'whose primary aim was to assess...'. However, I admit the authors would then probably have very little data to work with. As it is, please discuss the problem.

	C2) Be this as it may, there is nothing about how the controls were chosen. This part of PRISMA item 6 is completely disregarded in the text. If the specificity had not been uniformly high, that would have been a major criticism and would probably have led to scrapping of the meta-analysis. Personally, I can imagine at least three realistic and relevant types of control materials, quite apart from questions of early vs. later-phase disease: (a) suspected, disconfirmed cases of CD, (b) cases of diarrheal disease of unconfirmed nature, (c) proven cases of colitis ulcerosa – along with more or less well characterized mixtures of these. (Gastroenterologists may add refinements to this, but that would not detract from my methodological point.) Why is this issue left unmentioned? What is the clinical decision task envisaged? C3) At one point, the manuscript contains a break in its logic: The gastroenterologist’s problem is delineated in the first sentences of the Discussion (p8/18). It is said that CD symptoms and manifestations are all non-specific. In other words, the gastroenterologist’s problem is that he wants a specific test. False positives are his nightmare. Anti-GP2 comes onto the stage. It is more specific than hoped for (regardless of the precise delimitation of non-CD population, see C2). But the authors turn their thumbs down – because of poor sensitivity. My reaction is: if it is true that a specific test is sorely needed, then a non-invasive and not too expensive test like anti-GP2 which is rarely false positive and establishes the diagnosis in 1 of 5 or 6 cases of CD should be a welcome thing. So, in a sense, I would arrive at the opposite conclusion of the one in the manuscript. Language L1) P2/15: delete the second ‘were.’ P3/15: ‘is’ -> ‘are.’ P5/49: ‘illustrated’ -> ‘given’ (the process is illustrated, the chart that illustrates it is given/presented). P5/51: ‘full-text.’ L2) In the Discussion: On p8/24 please write ‘... is considered ... important factor in the diagnosis of...’ or ‘... is emerging as an important factor’ P8/46&54 & p9/4: ‘[un]satisfied’ -> ‘[un]satisfactory.’ P9/18: ‘owned ...’ -> ‘had more diagnostic value [in suspected CD]’ or the like. P9/28: ‘alone.’ P9/46: ‘researches’ -> either ‘investigations’ or singular ‘research.’ L3) P9/38-40: bias is investigated/looked for/tested (not performed), and the correlation is calculated/examined/tested (not performed). Please rephrase. ---//---
--	---

REVIEWER	Chelle Lorraine Wheat VA Puget Sound Health Care System Health Services Research & Development
-----------------	--

	1660 S Columbian Way S-152 Seattle WA 98108 United States
REVIEW RETURNED	18-Jan-2017

GENERAL COMMENTS	This paper reports the results of a systematic review and meta-analysis of the literature in order to evaluate the diagnostic value of the anti-glycoprotein 2 (GP2) antibody for Crohn's Disease. As the differentiation of Crohn's Disease (CD) and Ulcerative Colitis (UC) is a significant clinical decision, especially in regards to treatment and long term risk assessment, this manuscript provides an important contribution to the literature. I have been asked to specifically comment on the methodological quality of the manuscript. The primary strengths of this study include:  1) The novelty of the use of a systematic review and meta-analysis to evaluate this knowledge area. 2) The adherence of the study to the PRISMA guidelines. 3) The quality of the included studies to the review/meta-analysis assessed by QUADAS criteria. The primary weaknesses of the study include:  1) The specific search algorithms used to identify studies were not included. Without these, it is difficult to determine if a thorough search of the literature was conducted. Furthermore, it is unclear as to why conference abstracts were excluded from the analysis. By excluding the abstracts, unnecessary bias may have been introduced to the analysis. 2) The authors use the terms heterogeneity and inconsistency interchangeably which is confusing for readers who are less familiar with these ideas. 3) It is unclear as to why both random effects and fixed effects models are described. It appears that only random effects models were used in the analyses. I would recommend only including the models that were actually used. 4) The threshold effect should be explained more thoroughly for readers who are unfamiliar with the idea. 5) It is not immediately clear which studies were used for which analyses and why multiple meta-analyses were conducted. 5) Given the extremely high levels of heterogeneity, I question the decision to pool the studies. At a minimum, the interpretation of the results that the anti-GP2 is not a useful diagnostic tool is not supported by the analysis. 6) What were the sample sizes for each of the meta-analyses? In general, more detailed discussion of the methods is needed to adequately assess the methodological quality of the meta-analyses. Minor comments:  1) The text on page 5, line 46 that details the literature search does not match the information presented in Figure 1. Additionally, the other reasons for exclusion of the 19 full-text articles should be detailed. 2) Headers are missing on all pages except page 1 of table 1. I would recommend the following improvements:
---

	1) Provide a more detailed description of the methods used for the meta-analysis including a thorough explanation of heterogeneity, inconsistency, and the threshold effect. In addition, fully explain what models were used for which analyses and why the decision was made to pool the studies, and how readers should interpret these findings.
--	--

VERSION 1 – AUTHOR RESPONSE

Reviewer: 1

Comment 1. The importance of this study is not stated. Why was this study embarked upon? A paragraph discussing in detail what is known would be useful.

Response to comment 1: We appreciate your comment. We have revised the Introduction according to your suggestion (Introduction, Paragraph 3).

Reviewer: 2

Thank you for your positive review of our research and constructive comments and criticisms, which have helped us to significantly improve the quality of our manuscript.

Major points:

Comment 1. The conclusion of the meta-analysis should be revised. The novel anti-GP2 was reported as a specific marker of CD recently, which was corroborated by this meta-analysis. Anti-GP2 is associated with the clinical phenotype of CD (stricturing/stenosing) or even a putative subtype of CD. Thus, the sensitivity of around 20% cannot be used to postulate that anti-GP2 is “not a useful diagnostic marker for CD”.

Response to comment 1: Thank you for your professional comment. We have revised the conclusion according to your suggestion.

Comment 2. The association of the loss of tolerance to GP2 with the phenotype of disease in accordance with the Montreal classification should be mentioned in the INTRODUCTION section and discussed in the DISCUSSION section in line with the obtained meta-analysis data.

Response to comment 2: We appreciate your comment. However, most of the recruited studies did not provide the detection ratio of the anti-GP2 antibody among different subtypes of CD, so we were unable to perform the appropriate meta-analysis. Combining your suggestion and the current status of this study, we have included a detailed discussion of this point in the Discussion section, along with a brief mention in the Introduction section.

Comment 3. The possibility of the existence of different subtypes of CD should be elaborated in the DISCUSSION section in addition to the previous comment.

Response to comment 3: We agree with your comment, and we have added this in the revised manuscript per the previous comment.

Comment 4. The area under the curve (AUC) of a receiver operating characteristic (ROC) analysis indicates the probability that a randomly selected individual from the positive group has a test result indicating greater suspicion than that for a randomly chosen individual from the negative group. Thus, it is a measure of the assay accuracy. When the variable under study cannot distinguish between the two groups, i.e. where there is no difference between the two distributions, the area will be equal to 0.5. However, the AUC values of the summary ROC obtained in the paper seem to be higher than 0.5. Thus, the significance of the difference from 0.5 should be calculated for each AUC value given in the manuscript. Further, you cannot conclude that an AUC of 0.72 renders the parameter under investigation not useful unless the difference to 0.5 is not significant.

Response to comment 4: We agree with your comment. In the revised manuscript, we have added a 95% confidence interval (CI) to each AUC, as the software used for meta-analysis does not provide P-values. Results showed that the 95% CIs did not include 0.5, which implies that the AUC values are significantly different from 0.5. In addition, we have revised the Discussion section to avoid incorrectly concluding that the anti-GP2 antibody is not useful for diagnosis.

Comment 5. The meta-analysis revealed positive likelihood ratios of above 5 for anti-GP2. This means, that anti-GP2 has at least a 30% change in probability and, thus, at least a moderate effect on posttest probability of CD. Thus, anti-GP2 can be used for the differentiation of CD patients from controls, which should be concluded in the DISCUSSION section.

Response to comment 5: We agree with your comment, and we have added this conclusion to the Discussion section (Discussion, Paragraph 7).

Minor points:

Comment 1. ABSTRACT section, lines 12 - 17: "There were .." – This sentence needs revision on grammar.

Response to comment 1: Thank you for your suggestion. We have had our manuscript thoroughly reviewed for English language/usage by a professional editing service so as to enhance the readability of our manuscript.

Comment 2. RESULTS/MULTIPLE ... section, lines 40 – 49: The whole paragraph is difficult to comprehend and needs revision.

Response to comment 2: Thank you for your suggestion. We have had our manuscript thoroughly reviewed for English language/usage by a professional editing service so as to enhance the readability of our manuscript.

Comment 3. DISCUSSION section, line 21: Not all diagnostic methods indicated are invasive. Revise the sentence and correct the grammar.

Response to comment 3: Thank you for your professional suggestion. We have revised this sentence and corrected the grammar.

Comment 4. DISCUSSION section, line 35: Use the term "diagnostic performance"!

Response to comment 4: Thank you for your professional suggestion. We have revised this according to your suggestion.

Comment 5. Fig. 2, 3, 4 legends: Indicate the full term for SROC.

Response to comment 5: We have indicated the full term for SROC in these legends.

Comment 6. Fig. 5: Introduce the abbreviation "ESS" in the legend.

Response to comment 6: We have introduced the abbreviation "ESS" in the legend.

Comment 7. Suppl. Table 1: Correct "anti-granule protein 2" in the headline.

Response to comment 7: We have corrected this as "anti-glycoprotein 2 antibody".

Reviewer: 3

Comment 1. The Editors should make sure that figures are readable despite the small print! Authors - please see attachment.

Response to comment 1: Thank you for your suggestion. Since the number of eligible studies is more than 15, the current quality of the figures is the best that we can provide. If this quality is not acceptable for the journal, we could provide these as supplementary files in order to ensure their legibility.

Reviewer: 4

Thank you for your positive review of our research and constructive comments and criticisms, which have helped us to significantly improve the quality of our manuscript.

Major points:

Comment 1. The specific search algorithms used to identify studies were not included. Without these, it is difficult to determine if a thorough search of the literature was conducted. Furthermore, it is unclear as to why conference abstracts were excluded from the analysis. By excluding the abstracts, unnecessary bias may have been introduced to the analysis.

Response to comment 1: Thank you for your professional suggestion. Combined with a suggestion proposed by the editor, we have provided the specific search algorithm used for PubMed as Supplementary File 1. We agree with you that conference abstracts should not be excluded from the meta-analysis. We have rechecked all studies discovered in our search and determined that no

eligible conference abstracts were found.

Comment 2. The authors use the terms heterogeneity and inconsistency interchangeably which is confusing for readers who are less familiar with these ideas.

Response to comment 2: Thank you for this comment. We have replaced the term inconsistency with the term heterogeneity, as this is a more descriptive expression. We have also added an explanation of the definition of heterogeneity to the Methods section.

Comment 3. It is unclear as to why both random effects and fixed effects models are described. It appears that only random effects models were used in the analyses. I would recommend only including the models that were actually used.

Response to comment 3:

Generally, applying which effects model in the meta-analysis is mainly based on the the degree of the heterogeneity. Before we calculate the heterogeneity, we can not be sure that which effect model will be used in our research. So that we describe both random effects and fixed effects models in the Methods section.

Comment 4. The threshold effect should be explained more thoroughly for readers who are unfamiliar with the idea.

Response to comment 4: Thank you for this suggestion. We have added an explanation of the threshold effect to the Methods subsection Statistical analysis.

Comment 5. It is not immediately clear which studies were used for which analyses and why multiple meta-analyses were conducted.

Response to comment 5: We agree with your comment. Since previous reports showed that different subtypes of anti-GP2 antibody provided different diagnostic accuracies for CD, we conducted multiple meta-analyses to assess differences in diagnostic performance among them. We have added a related description to the Introduction section and references to each meta-analysis (Results section) so that the reader can immediately discern which studies were used for which analyses.

Comment 6. Given the extremely high levels of heterogeneity, I question the decision to pool the studies. At a minimum, the interpretation of the results that the anti-GP2 is not a useful diagnostic tool is not supported by the analysis.

Response to comment 6: In the meta-analysis evaluating the diagnostic index, studies can be pooled together when there are no threshold effects. Our research did not find any threshold effects, so we chose to calculate the pooled sensitivity and specificity. We also agree that our conclusion was inappropriately drawn, as there may exist other sources of heterogeneity not investigated. We have revised the Discussion thoroughly to draw a conclusion supported by the analysis.

Comment 7. What were the sample sizes for each of the meta-analyses? In general, more detailed discussion of the methods is needed to adequately assess the methodological quality of the meta-analyses.

Response to comment 7: We appreciate your comment. We have tried to add more discussion of the methods to the Results and Discussion section. In addition, we have added the sample sizes for each of the meta-analyses to the Results section.

Minor points:

Comment 1. The text on page 5, line 46 that details the literature search does not match the information presented in Figure 1. Additionally, the other reasons for exclusion of the 19 full-text articles should be detailed.

Response to comment 1: We apologize for this error. We have revised the description in the Results subsection Literature search. The reason that the 19 full-text articles were excluded is that they were not related to our subject, which has been added to the revised manuscript.

Comment 2. Headers are missing on all pages except page 1 of table 1.

Response to comment 2: Thank you for your comment. We have corrected this.

Recommend improvements:

Comment 1. Provide a more detailed description of the methods used for the meta-analysis including a thorough explanation of heterogeneity, inconsistency, and the threshold effect. In addition, fully explain what models were used for which analyses and why the decision was made to pool the studies, and how readers should interpret these findings.

Response to comment 1: We have made relevant revisions according to the major and minor comments that you have provided. Thank you again for your constructive suggestions, which have improved our manuscript substantially.

Our point-by-point responses to the editorial requirements:

Requirement 1. Please revise the discussion and conclusion section in accordance with the comments of the Reviewers, to present a more cautious overview of the results.

Response to requirement 1: Thank you for this suggestion. We have revised the Discussion and Conclusion sections according to the comments of the reviewers.

Requirement 2. Please include an 'Article summary' section consisting of the heading: 'Strengths and limitations of this study', and containing up to five short bullet points, no longer than one sentence each, that relate specifically to the methods of the study reported. This should be placed after the abstract.

Response to requirement 2: We have added the article summary according to your instructions.

Requirement 3. Please include the full search strategy for at least one database as a supplementary file.

Response to requirement 3: We have provided our exact search strategy for PubMed as Supplementary File 1.

VERSION 2 – REVIEW

REVIEWER	Dirk Roggenbuck Institute of Biotechnology, Brandenburg University of Technology Cottbus-Senftenberg, Germany Shareholder of Medipan and GA Generic Assays GmbH
REVIEW RETURNED	21-Feb-2017

GENERAL COMMENTS	The authors have addressed all comments raised appropriately.
---

REVIEWER	Jørgen Hilden University of Copenhagen, Denmark
REVIEW RETURNED	24-Feb-2017

GENERAL COMMENTS	Review of Deng, Li & al., “Diagnostic value of anti-glycoprotein-2 antibody for Crohn’s disease ... “ BMJ-open-2016-014843, Revised R1 The authors have done a lot to improve language, clarity and structure. I still see some weaknesses, however. References are to px/y authors’ page no. x, pdf line no. y. X1) It should be stressed from the outset what is meant by the ‘either’ phrase (‘either IgG or IgA’). It means that ‘either IgG or IgA is found to be present (abnormal).’ Formally, the combined test is taken as positive if one or both component tests is positive (logical OR of positive results); it is negative if both components are negative (logical AND of negativity). Please clarify. – Regardless how this is done, the positivity criterion (logical OR of positivity) is
--

misrepresented on p11/15. In fact, the phrase 'detecting both the IgG and IgA subtype' suggests that a logical AND is employed (combined test positive only if both subtypes are detected). So the word 'detecting' should be replaced with 'looking for,' 'testing for' or something similar.

X2) Ref. [4] is identical to ref. [21].

X3) A Spearman correlation is calculated but the reader is not told between which variables. I presume it is a correlation between sens and (1 – spec) across source studies.

X4) In fact, a fairly strong and significant correlation is found in each of the three cases (p9/32-46). To me this looks as evidence in favour of a threshold or threshold-imitating (e.g., severity or chronicity) effect, but the text takes the calculations as evidence against such effect. Please clarify. The paragraph as a whole is difficult to read.

X5) Concerning Figs. 2-4: Isn't it so that the actual meta-analytic result shown in part C of the figures is only given via a diamond at the central point in the graph (with no mention of its coordinates, and without these coordinates being quoted in the text)? – Also, the prediction region (outer curve) of part C requires a small explanation in order to distinguish it from the confidence region for the central point.

X6) The PRISMA checklist assumes that the source studies were undertaken precisely in order to answer the question whether a test is useful for the diagnostic task at hand; in other words, they should have the same aim as the meta-analysis. However, going through the list of refs., I find many titles that suggest that the diagnostic 2-by-2 table was just a by-product of an investigation with a different aim. Such datasets are possibly not very concerned about representative samples, etc., and have been subject to selective forces in the publication process that may bias the meta-analysis in unpredictable ways. To be specific, three titles concern subtypes of CD [7, 13, 18]; two have attention on refractory cases [14] or development of tolerance [11]; three appear to focus on molecular processes [15, 17, 22], and one advertises a new kit [10]. Novelty is a theme in [21] ('exciting-news bias?'), and the purpose of [16] is to warn against a diagnostic 'trap' (publication prompted by disappointing results?). Under such circumstances, heterogeneity is hardly surprising.

X7) Be this as it may, there is nothing about how the so-called controls were chosen. This part of PRISMA item 6 is completely disregarded. If the specificity had not been uniformly high, that would have been a major criticism and would probably have led to scrapping of the meta-analysis. What symptoms, history and prior diagnoses were the so-called controls required having? We agree that they were not just healthy people from the street (had they been, the meta-analysis would have been useless). They were people with complaints that made Crohn a relevant diagnostic possibility, of course. But the reader gets not details. Why is this issue left unmentioned?

X8) The gastroenterologist's problem is delineated early on. It is said that CD symptoms and manifestations are all non-specific. In other words, the gastroenterologist's problem is that he wants a specific test. False positives are his nightmare. Now anti-GP2 turns out to be

	very specific (regardless of the precise delimitation of non-CD population). Shouldn't this asset be stressed a little more strongly? X9) In particular, I feel the text takes a wrong turn in the Discussion (p10/53), when it concludes that the high specificity, combined with a fairly low sensitivity, suggests that the antibody test 'may identify controls better than patients with CD.' What does it mean to identify a medical condition? To 'identify' the condition in a given patient means to conclude, with little risk of error, that the condition is present, so one requires a test result that excludes, or comes close to excluding, rival conditions. To identify the presence of CD one requires a test result that rarely suggests CD when CD is absent (there are few false positives; specificity is high). To identify the absence of CD one requires a test result that rarely suggests absence of CD when in fact it is present (few false negatives; sensitivity high). So, in the high specificity-low sensitivity situation we are dealing with, CD is more reliably identified than non-CD. (Admittedly, this occurs at the price of allowing identification of only a small fraction of CD cases because so the majority are false negatives.) X10) Are differences between manufacturers a cause of heterogeneity? A positive answer is given at p9/29 and again at p11/33. However, p11/26 says the opposite (the language editor has probably overlooked a negation?). ---***---
--	---

REVIEWER	Chelle Lorraine Wheat Veterans Health Administration - Puget Sound United States
REVIEW RETURNED	06-Mar-2017

GENERAL COMMENTS	The revised manuscript is much improved in regards to the overall writing and description of the methods. I believe that it is now acceptable for publication if agreed upon by the editor. One minor suggestion for additional improvement is in regards to Figure 1. I suggest moving the boxes with the excluded records above the records screened so that the flow is logical.
---

VERSION 2 – AUTHOR RESPONSE

Reviewer: 3

Thank you again for your positive review of our research and constructive comments and criticisms, which have helped us to significantly improve the quality of our manuscript.

Comment 1. It should be stressed from the outset what is meant by the 'either' phrase ('either IgG or IgA'). It means that 'either IgG or IgA is found to be present (abnormal).' Formally, the combined test is taken as positive if one or both component tests is positive (logical OR of positive results); it is negative if both components are negative (logical AND of negativity). Please clarify. – Regardless how this is done, the positivity criterion (logical OR of positivity) is misrepresented on p11/15. In fact, the phrase 'detecting both the IgG and IgA subtype' suggests that a logical AND is employed (combined test positive only if both subtypes are detected). So the word 'detecting' should be replaced with 'looking for,' 'testing for' or something similar.

Response to comment 1: We totally agree with you comments. We have clarified the 'either IgG or

IgA' phrase in the Method, Data extraction. In addition, we have replaced the word 'detecting' with 'testing for' (Discussion paragraph 6).

Comment 2. Ref. [4] is identical to ref. [21].

Response to comment 2: We apologize for the mistake and have revised that.

Comment 3. A Spearman correlation is calculated but the reader is not told between which variables. I presume it is a correlation between the sens and (1 – spec) across source studies.

Response to comment 3: Thank you for you reminding. The spearman correlation is calculated between the logarithm of sensitivity and the logarithm of (1 – specificity) across source studies. We have described that in the Methods, Statistical analysis.

Comment 4. In fact, a fairly strong and significant correlation is found in each of the three cases (p9/32-46). To me this looks as evidence in favour of a threshold or threshold-imitating (e.g., severity or chronicity) effect, but the text takes the calculations as evidence against such effect. Please clarify. The paragraph as a whole is difficult to read.

Response to comment 4: We have checked and confirmed that there is threshold effect. We apologized for the mistake and have rewritten the whole paragraph to make it more readability.

Comment 5. Concerning Figs. 2-4: Isn't it so that the actual meta-analytic result shown in part C of the figures is only given via a diamond at the central point in the graph (with no mention of its coordinates, and without these coordinates being quoted in the text)? – Also, the prediction region (outer curve) of part C requires a small explanation in order to distinguish it from the confidence region for the central point.

Response to comment 5: Thank you for your suggestion. We have added more description or discussion of the generated SROCs, including the 95% prediction contour and the 95% confidence contour in the SROCs (Methods, Statistical analysis; Results, Meta-analysis of the diagnostic accuracy of the anti-GP2 antibody for CD).

Comment 6. The PRISMA checklist assumes that the source studies were undertaken precisely in order to answer the question whether a test is useful for the diagnostic task at hand; in other words, they should have the same aim as the meta-analysis. However, going through the list of refs., I find many titles that suggest that the diagnostic 2-by-2 table was just a by-product of an investigation with a different aim. Such datasets are possibly not very concerned about representative samples, etc., and have been subject to selective forces in the publication process that may bias the meta-analysis in unpredictable ways. To be specific, three titles concern subtypes of CD [7, 13, 18]; two have attention on refractory cases [14] or development of tolerance [11]; three appear to focus on molecular processes [15, 17, 22], and one advertises a new kit [10]. Novelty is a theme in [21] ('exciting-news bias?'), and the purpose of [16] is to warn against a diagnostic 'trap' (publication prompted by disappointing results?). Under such circumstances, heterogeneity is hardly surprising.

Response to comment 6: Thank you for your professional comment. We agree that the recruited studies should have the same aim as the meta-analysis. Sometimes, few studies were undertaken precisely to confirm a test is useful for the diagnostic task or not, such as the test of anti-GP2 antibody. In order to evaluate this clinical index with potential value early, we selected the current studies. Previously, we have conducted the evaluation of quality among these articles by using QUADAS tool, so that the readers can have a preliminary understanding of the diagnostic value of these articles. Your comment reminds us that this is not enough, and we have illustrated your valuable comments in the limitation part, so that the readers can further understand recruiting these articles in the meta-analysis may introduce potential heterogeneity (Discussion, paragraph 10).

Comment 7. Be this as it may, there is nothing about how the so-called controls were chosen. This part of PRISMA item 6 is completely disregarded. If the specificity had not been uniformly high, that would have been a major criticism and would probably have led to scrapping of the meta-analysis. What symptoms, history and prior diagnoses were the so-called controls required having? We agree that they were not just healthy people from the street (had they been, the meta-analysis would have been useless). They were people with complaints that made Crohn a relevant diagnostic possibility, of course. But the reader gets not details. Why is this issue left unmentioned?

Response to comment 7: We are deeply sorry for missing this part because of our negligence. We

only included the studies that recruiting people with complaints (abdominal pain, diarrhea, and ileus, etc.) that made CD a relevant diagnostic possibility, or patients diagnosed with the diseases that need to be differential with CD, such as ulcerative colitis, irritable bowel syndrome, ischemic bowel disease and so on, as controls. We have added that to the manuscript (Methods, Eligibility criteria).

Comment 8. The gastroenterologist's problem is delineated early on. It is said that CD symptoms and manifestations are all non-specific. In other words, the gastroenterologist's problem is that he wants a specific test. False positives are his nightmare. Now anti-GP2 turns out to be very specific (regardless of the precise delimitation of non-CD population). Shouldn't this asset be stressed a little more strongly?

Response to comment 8: We appreciate this comment. We have stressed this asset in the discussion according to your suggestion (Discussion, 5).

Comment 9. In particular, I feel the text takes a wrong turn in the Discussion (p10/53), when it concludes that the high specificity, combined with a fairly low sensitivity, suggests that the antibody test 'may identify controls better than patients with CD.' What does it mean to identify a medical condition? To 'identify' the condition in a given patient means to conclude, with little risk of error, that the condition is present, so one requires a test result that excludes, or comes close to excluding, rival conditions. To identify the presence of CD one requires a test result that rarely suggests CD when CD is absent (there are few false positives; specificity is high). To identify the absence of CD one requires a test result that rarely suggests absence of CD when in fact it is present (few false negatives; sensitivity high). So, in the high specificity-low sensitivity situation we are dealing with, CD is more reliably identified than non-CD. (Admittedly, this occurs at the price of allowing identification of only a small fraction of CD cases because so the majority are false negatives.)

Response to comment 9: Thank you this perspective and we agree. We have revised that according to your suggestion (Discussion, 5).

Comment 10. Are differences between manufacturers a cause of heterogeneity? A positive answer is given at p9/29 and again at p11/33. However, p11/26 says the opposite (the language editor has probably overlooked a negation?).

Response to comment 10: Thank you for your comment. We have checked and confirmed that there might be misunderstanding between the method of autoantibody detection (indirect immunofluorescence and enzyme-linked immuno sorbent assay) and the manufacturer of detection kits (Euroimmune, GA Generic, Inova Diagnostics or in-house kits). We have defined that in the Results (Multiple regression analysis and exploration of threshold effect) to avoid the misunderstanding.

Reviewer: 4

Thank you for your positive review of our research and constructive comments and criticisms, which have helped us to significantly improve the quality of our manuscript.

Comment 1. One minor suggestion for additional improvement is in regards to Figure 1. I suggest moving the boxes with the excluded records above the records screened so that the flow is logical.

Response to comment 1: Thank you for your kind comment and we have revised the figure according to your suggestion.

VERSION 3 – REVIEW

REVIEWER	Jørgen Hilden University of Copenhagen, Biostatistics, Denmark
REVIEW RETURNED	20-Apr-2017

GENERAL COMMENTS	Review of manuscript by Deng, Li & al., "Diagnostic value of anti-glycoprotein-2 antibody for Crohn's disease ... " BMJ-open-2016-014843, Revision R2
--

	The authors have reacted nicely in my opinion to the comments made by the reviewer group. The text absolutely publishable and topical. However, I still worry a bit about the small print in some of the figures.
--	--